# Molecular Landscape of Prostate Cancer Across Age Groups: Impact on Prognosis and Treatment Outcomes

**DOI:** 10.3390/ijms26199777

**Published:** 2025-10-08

**Authors:** Magdalena Julita Orzechowska, Andrzej K. Bednarek

**Affiliations:** Department of Molecular Carcinogenesis, Medical University of Łódź, 90-752 Łódź, Poland

**Keywords:** neoplasm grading, precision medicine, tumor microenvironment, prostatic neoplasms, early-onset prostate cancer, late-onset prostate cancer, aged

## Abstract

Prostate cancer (PC) has long been considered a disease of older men. Still, a significant and concerning rise in diagnoses among younger men has revealed a biologically distinct and more aggressive clinical entity: early-onset prostate cancer (EO-PC). This comprehensive review synthesizes the molecular and clinical evidence to demonstrate that PC is not a single disease, but a collection of distinct entities delineated by patient age. EO-PC is characterized by a strong genetic component, unique fusion events like *TMPRSS2-ERG*, and a highly plastic phenotype driven by intense Notch signaling and a hybrid epithelial-to-mesenchymal transition. In stark contrast, late-onset prostate cancer (LO-PC) is defined by a higher mutational burden, an epigenetic “field defect” that accumulates with age, and a predominantly immunosuppressive tumor microenvironment. These profound biological differences have significant implications for diagnosis, prognosis, and therapeutic strategies. Traditional prognostic tools, such as the Gleason score, are often insufficient to capture the full spectrum of risk in younger men. The divergent molecular landscapes of EO-PC and LO-PC necessitate a fundamental shift from a standard approach to an age-aware precision medicine framework. This review highlights key therapeutic targets and underscores the critical need for a new paradigm in PC management to improve patient outcomes.

## 1. Introduction

Prostate cancer (PC) stands as one of the most common malignancies diagnosed in men globally, posing a profound and accelerating burden on public health. Recent epidemiological data highlight a significant increase, with over 10 million cases recorded worldwide in 2021, representing a nearly 189% rise since 1990 [1]. Projections indicate a near-doubling of annual new cases from 1.8 million to 2.9 million between 2020 and 2040 [2]. This alarming trajectory is predominantly driven by two demographic factors: population growth, accounting for 65.62% of the increase, and the global aging of the populace, which contributes a significant 16.41% to the rising case numbers [1].

In this report, Early-Onset Prostate Cancer (EO-PC) is defined as a diagnosis established at age ≤ 55 years, while Late-Onset Prostate Cancer (LO-PC) includes men diagnosed at age ≥ 65 years. The above age distinction is not arbitrary but reflects fundamentally different evolutionary pathways of the tumor [3,4,5,6].

While the incidence of PC has risen, a significant decline in the mortality rate has been observed in high-income countries over the past three decades. This is a testament to the widespread adoption of prostate-specific antigen (PSA) screening and advances in multimodal therapies. Conversely, low- and middle-income countries face persistently elevated case fatality rates. This disparity highlights systemic inequities in healthcare infrastructure and access to early detection and treatment innovations [2].

As highlighted in the comprehensive Cancer Statistics by Siegel et al., significant racial disparities in PC mortality persist; for instance, the mortality rate among Black men (36.9 deaths per 100,000) is more than double that of White men (18.4 deaths per 100,000) [7].

This epidemiological landscape underscores a critical need to re-evaluate the global burden of PC through a framework that integrates demographic transitions with the underlying biological risk factors and molecular drivers of the disease. The association between advancing age and PC incidence is striking, with the highest relative risk concentrated around age 75 and over 60% of cases diagnosed in men aged 65 or older. This strong clinical connection suggests that PC is not a monolithic disease entity but a collection of distinct biological conditions. The above necessitates a transition from a standard approach to an age-aware precision medicine framework that acknowledges age as a fundamental differentiating factor. This review details the divergent molecular and clinical trajectories that distinguish EO-PC from its late-onset counterpart, providing a direct comparison necessary for developing age-aware precision medicine strategies.

The underlying mechanisms driving age-related PC are rooted in fundamental biological processes that occur at the cellular and tissue level of aging itself [1]. To understand the molecular genesis of age-related diseases like PC, one must first appreciate the biological underpinnings of aging per se. The seminal work of López-Otín et al. provides a foundational framework by identifying twelve interconnected hallmarks of aging that collectively drive functional decline with age. These hallmarks, including genomic instability, telomere attrition, epigenetic alterations, loss of proteostasis, disabled macroautophagy, deregulated nutrient-sensing, mitochondrial dysfunction, cellular senescence, stem cell exhaustion, altered intercellular communication, and chronic inflammation, are not isolated phenomena. Instead, they are deeply interconnected in a complex network of cause and effect, where a disturbance in one hallmark can cascade to affect others. This creates a permissive milieu for malignant transformation, mechanistically explaining the striking association between advancing age and the incidence of most cancers, including PC. In the context of LO-PC, the tumor is the result of a gradual, stochastic accumulation of damage stemming directly from these hallmarks of aging. Conversely, in EO-PC, the malignancy must result from acute, highly penetrant genetic factors or catastrophic genomic events that bypass the need for decades of damage accumulation, resulting in two fundamentally different biologies and clinical courses [8].

## 2. Clinical and Pathological Characteristics

### 2.1. Incidence and Epidemiology: A Rising Tide in Younger Men

The incidence of PC is strongly correlated with age, with a rapid increase in diagnosis rates in men over the age of 50. Despite this well-known trend, the number of individuals diagnosed below the age of 55 has been rising annually, drawing increased clinical attention to this younger patient population. Three primary, well-established risk factors for PC include increasing age, a family history of the disease, and Black ethnicity. Notably, the demographic shift towards earlier-onset disease is particularly pronounced in men of African ancestry, who have a higher lifetime risk of developing PC (1 in 4) compared to white men (1 in 8) and are more likely to be diagnosed at a younger age. These epidemiological patterns hint at a strong genetic component driving the early onset of the disease, a factor that is also amplified by a strong family history. The risk of developing PC rises significantly with the number of affected relatives, and if those relatives were diagnosed at a younger age, with a first-degree relative diagnosed before age 65, increasing an individual’s risk nearly threefold [9].

A recent large-scale, population-based cohort study further quantified this familial risk by examining PC characteristics in fathers and the risk of early-onset disease in their sons. The study, which included 25,287 men in Sweden whose fathers had been diagnosed with PC, found that their overall risk of a PC diagnosis was more than tripled compared to the expected incidence in the Swedish male population (SIR, 3.2; 95% CI, 2.9–3.4). This overall standardized incidence ratio (SIR) is greater than previously reported, with a meta-analysis estimating a relative risk of 2.35 (95% CI, 2.02–2.72). The higher risk observed in this study is likely a reflection of the selection of an early-onset disease cohort, as the median age of the sons at the end of follow-up was only 52 years, suggesting that these results are specific to early-onset cases [10].

The study provided critical, nuanced data on how a father’s age at diagnosis influences the risk in his sons. The sons’ overall risk of PC was found to be highest if the father had been diagnosed at less than 65 years old, conferring a four-fold higher overall incidence than expected (SIR, 4.3; 95% CI, 3.8–5.0). The SIRs gradually decreased as the father’s age at diagnosis increased, with a father diagnosed at 70 years old or older conferring a lower, though still elevated, risk to his sons (SIR, 2.3; 95% CI, 1.9–2.8) [10].

### 2.2. Disease Presentation and Prognosis: The Clinical Paradox

The clinical presentation of EO-PC reveals a complex and seemingly contradictory picture, as summarized in Table 1. A substantial portion of men aged 55 or younger are diagnosed with low-risk disease. This is reflected in high overall survival rates, with men aged 20–54 having a 5-year relative survival rate of 98.0%, which is only slightly lower than the 100.0% survival rate for men aged 55–79 diagnosed in the same period [11]. However, this benign-appearing statistic masks a critical underlying reality.

A subset of young men with high-grade (Gleason score 8–10) or advanced-stage prostate cancer faces an inferior prognosis [12]. Among men diagnosed with high-grade and high-stage disease, those with EO-PC are more likely to die from their cancer, exhibiting a higher cause-specific mortality than all other age groups, except those diagnosed over 80. For example, a study of men diagnosed between 1988 and 2003 found that those aged 35–44 with stage IV cancer had an approximately 1.5-fold greater risk of dying from their cancer compared to men aged 65–74 [11].

This paradoxical presentation suggests that the “early-onset” diagnosis encompasses two distinct disease entities. The large number of low-risk cases may be an artifact of increased PSA screening in genetically predisposed populations, capturing indolent disease that may never become clinically significant. It is further explained by a direct concordance of tumor characteristics between generations, as aforementioned in population-based studies. Lin et al. found that having a father with a high Gleason score cancer did not confer a higher overall risk for a PC diagnosis in the son compared with a low Gleason score. However, a high Gleason score (≥8) in the father was associated with a higher risk for a similar high-Gleason cancer in the son (SIR, 2.6; 95% CI, 1.1–5.1), an association that was stronger than that conferred by a father with a low Gleason score (SIR, 1.6; 95% CI, 1.0–2.6). These findings suggest that the inherited risk is not merely for PC in general, but for a specific, aggressive subtype. The heritability of disease aggressiveness is further demonstrated by a combined analysis, which found that the combination of an early onset and a high Gleason score in the father’s cancer was associated with a particularly increased risk of high-Gleason cancer in the son (SIR, 4.7; 95% CI, 0.94–13.7), although these estimates had low precision due to a limited number of events [10]. This empirical data reinforces the understanding that the high-grade, lethal cases observed in younger men are likely driven by a biologically aggressive, genetically distinct subtype from the outset.

Given that EO-PC is deeply rooted in genetic factors, genetic testing becomes a pivotal diagnostic component, enabling the assessment of the true risk of disease progression and guiding more personalized therapeutic decisions, even in cases with a seemingly benign initial clinical picture [13].

### 2.3. Diagnostic Challenges and Long-Term Implications

One of the significant challenges in detecting EO-PC is the non-specific nature of its symptoms. Early-stage prostate cancer often causes no symptoms at all. When signs do appear, they can mimic those of benign prostatic hyperplasia (BPH), a non-cancerous growth common in older men. Symptoms may include a slow or weak urinary stream, frequent urination, or trouble emptying the bladder [14]. This non-specific presentation can lead to a missed or delayed diagnosis.

Furthermore, the standard PSA test, a cornerstone of PC diagnosis, is an unreliable indicator in some young men with aggressive disease. PSA is a glycoprotein secreted by prostate epithelial cells, and its levels can be elevated in both benign and malignant conditions [15]. However, in young men with poorly differentiated, highly aggressive carcinomas, the cancer cells may have lost their secretory function, resulting in a misleadingly low PSA level despite the presence of aggressive disease [12]. This creates a diagnostic pitfall where an aggressive cancer can be overlooked or diagnosed only at an advanced stage, underscoring a critical need for new biomarkers beyond PSA for early detection in high-risk young men, especially those with a strong family history or of African ancestry.

### 2.4. The Extended Burden of Treatment

Due to their extended life expectancy, young men with prostate cancer face a prolonged risk of disease progression and death. They also endure the long-term impact of treatment-related morbidities, which can last for decades. Treatments such as hormone therapy or androgen deprivation therapy (ADT), can lead to a host of significant side effects, including hot flashes, fatigue, decreased bone strength, weight gain, and low libido [11]. The management of these chronic conditions becomes a lifelong clinical challenge, requiring comprehensive care plans that extend far beyond the initial treatment phase. This emphasizes the need for careful consideration of the long-term quality of life when making treatment decisions for this patient population.

## 3. Molecular and Genetic Landscape

### 3.1. Divergent Evolutionary Trajectories: The Genomic Blueprint

The clinical and pathological differences observed in PC across age groups are not random; they are a direct consequence of profoundly different molecular and cellular programs. The multi-omic analysis of tumors from young and old patients reveals two distinct biological entities with divergent evolutionary trajectories, such as the so-called alternative evotype involving fusion-driven oncogenesis in EO-PC and the canonical evotype representing an accumulation of genomic damage in LO-PC [16].

EO-PC, typically defined as a diagnosis before the age of 55, is characterized by a higher frequency of specific fusion events that define its unique evotype. These include fusions involving the genes *TMPRSS2*, *ETV1*, *ETV4*, and *BRAF* [17]. The *TMPRSS2-ERG* fusion, in particular, is considered one of the earliest truncal drivers of prostate carcinogenesis and is found in approximately 40–50% of PC cases [18]. This fusion is a primary trigger in younger patients, leading to the activation of otherwise inactive, cancer-related genes, often linked to androgen signaling [16].

The formation of these fusions is often the result of a catastrophic genomic event known as chromoplexy [19,20]. This phenomenon, first identified in PC, is distinct from other forms of genomic chaos like chromothripsis [21]. Unlike chromothripsis, which affects a single chromosome with hundreds of locally clustered breakpoints, chromoplexy affects several chromosomes at once, creating complex, balanced translocations and “all-in-one” genome reshuffling events. This represents a burst of punctuated evolution, where a single event can generate multiple oncogenic advantages, providing a decisive proliferative advantage to a pre-cancerous cell [19]. For example, in at least one prostate tumor, a single chromoplectic event generated the *TMPRSS2-ERG* fusion while simultaneously inactivating key tumor suppressor genes like *SMAD4* and *PTEN*. This is a decisive distinction, i.e., the genomic instability in EO-PC is not a slow, stochastic accumulation of damage, but a burst-like, catastrophic event that defines the aggressive nature of the disease from the outset [21].

In stark contrast to EO-PC, LO-PC, diagnosed in older men (≥65 years), exhibits a different genomic signature that reflects a canonical-evotype [22,23]. While fusions are less frequent, LO-PC tumors have a higher overall tumor mutational burden (TMB), a pattern dominated mainly by spontaneous C>T transitions. This “clock-like” mutational signature reflects the gradual, cumulative accumulation of genomic damage over a lifetime [24]. This slower, more stochastic evolutionary trajectory leads to a different profile of driver mutations, with older tumors showing a higher frequency of mutations in key tumor suppressor genes, including *APC*, *CTNNB1*, *RB1*, and *AR* [17]. The higher prevalence of *RB1* alterations in LO-PC is particularly significant, as this genetic feature is a known driver of resistance to CDK4/6 inhibitors in other cancer types and may confer a pre-existing resistance mechanism to specific therapies in older patients [25]. Furthermore, a higher rate of dMMR/MSI-H signatures is observed in LO-PC compared to EO-PC, a direct result of compromised DNA repair mechanisms that accompany aging [15].

### 3.2. Inherited Genetic Predisposition (Germline Mutations)

The high heritability of prostate cancer is well-established, with genetic factors accounting for up to 60% of the risk. This heritability is highest among all common cancers, with an observed 58% heritability in a Norwegian twin study. EO-PC is particularly associated with a strong genetic component, with a higher likelihood of constitutional (germline) genetic variants as the underlying cause [9].

Pathogenic variants in the DNA repair genes *BRCA1* and *BRCA2* are among the most significant high-risk genetic factors. Carriers of deleterious variants in *BRCA2* individuals face a particularly high risk, with a 7.5-fold higher probability of developing aggressive prostate cancer compared to those with non-aggressive disease. Furthermore, patients with *BRCA1/2* variants are more likely to have metastatic disease at the time of diagnosis and have a reduced survival time. The *HOXB13* G84E germline variant is another critical genetic marker, identified as the most common single-gene pathogenic variant associated with a strong family history of PC in men of European ancestry. The presence of this variant increases an individual’s lifetime risk of developing prostate cancer by fivefold [9].

There is a compelling interplay between genetics, ethnicity, and age that influences the risk profile. Men of African ancestry have a higher lifetime risk of PC and are diagnosed at a younger age than men of other ethnicities. This may be due, in part, to a higher polygenic risk score (PRS) in this population. The confluence of a higher inherited risk and a higher propensity for specific, aggressive somatic mutations, such as in *CDK12*, may explain both the higher incidence and earlier age of diagnosis in this population [9]. This synergistic risk model highlights the need for genetically and ethnically sensitive screening and counseling programs.

### 3.3. Acquired Somatic Alterations

Several distinct acquired somatic alterations also define the molecular landscape of EO-PC. A study using the American Association for Cancer Research Project Genomics Evidence Neoplasia Information Exchange (GENIE) registry found that EO-PC patients had a higher likelihood of having mutations in *CDK12* and *ERF* [26].

The *CDK12* gene encodes a cyclin-dependent kinase that regulates transcriptional elongation and is a master regulator of DNA damage response (DDR) pathways, including those involving *BRCA1* and *BRCA2*. Loss-of-function mutations in *CDK12* lead to genomic instability, which is a hallmark of aggressive cancers [27]. However, this very instability creates a specific therapeutic vulnerability. Cancer cells with a compromised DDR pathway become dependent on alternative DNA repair mechanisms. This dependency can be exploited through a concept known as so-called “synthetic lethality,” where a drug that inhibits the alternative repair pathway will cause selective apoptosis in the cancer cells without harming healthy cells [9].

The *ERF* gene acts as a tumor suppressor. Loss-of-function mutations in *ERF* are particularly common in early-onset tumors and are mutually exclusive with the *TMPRSS2-ERG* gene fusion, a well-known hallmark of many other PC cases. The loss of *ERF* function has been shown to recapitulate the oncogenic effects of *ERG* gain. It may serve as a predictive marker of a patient’s response to anti-androgen therapy, indicating a clear link between a specific somatic mutation and a potential therapeutic response [28].

### 3.4. The Epigenetic Clock and Transcriptional Reprogramming

The molecular divergence between age groups extends beyond the genome to the epigenetic and transcriptional levels, profoundly influencing the biology of the disease. The aging process is accompanied by specific, reproducible DNA methylation changes that contribute to a permissive environment for malignant transformation. These age-associated changes create what has been termed an “epigenetic field defect” or “Areas of Epigenetic Susceptibility” (AREAS). These are widespread regions of the prostate tissue that undergo epigenetic alterations, increasing the tissue’s susceptibility to develop cancer in older men. These changes accumulate over a lifetime due to endogenous and exogenous exposures such as oxidative stress and nutritional factors [29].

Specific epigenetic markers, such as focal hypermethylation at the promoters of genes like *GSTP1*, *RARβ2*, and *RASSF1A*, have been identified as key alterations common to both aging and cancer [29,30]. A study that evaluated the methylation status of these genes in 74 cases of localized PC found that *RASSF1A* was methylated in 100% of samples, *GSTP1* in 78.4%, and *RARβ2* in 73% [29]. The translational utility of this concept is demonstrated by diagnostic tests like ConfirmMDx, which relies on detecting hypermethylation of *GSTP1*, *APC*, and *RASSF1* in histologically benign biopsy cores to predict the presence of occult cancer [29,30,31,32,33].

Beyond the transcriptome, the epigenetic landscape also plays a crucial role. Epigenetic factors, such as DNA methylation and histone modifications, are sensitive to environmental influences and change with aging. Periods of rapid prostate growth, such as during in utero development and puberty, are identified as so-called windows of susceptibility for epigenetic alterations that may play a role in adult-onset disease progression [29].

## 4. A Comparative View on the Tumor Microenvironment and Cellular Plasticity

### 4.1. The Epithelial–Mesenchymal Transition (EMT)

The hallmark of cellular plasticity, particularly the process of epithelial-to-mesenchymal transition (EMT), is profoundly modulated by patient age, and the molecular mechanisms of invasiveness differ starkly between young and old patients [34]. In tumors from younger patients, a pattern of partial EMT is observed, characterized by the simultaneous expression of both epithelial (*CDH1*) and mesenchymal markers (*CTNNB1* and *FN1*) [34,35]. This hybrid phenotype is a key feature of tumor plasticity, allowing cells to migrate while retaining some epithelial characteristics [34]. The invasive behavior in this cohort is driven by cell migration facilitated by stress fiber contractions and detachment from the ECM. This process is supported by *MMP7* activity and involves focal adhesion via *LAMA1* [35].

Conversely, tumors in older patients exhibit a more intensified EMT program that is distinct from the hybrid phenotype of younger men. This different invasive phenotype is driven by a broader range of transcription factors, including *SNAI2*, *TWIST1*, *ZEB2*, *HIF1A*, and *SMAD3*. While late-stage tumors in older men can restore epithelial marker expression like *CDH1*, they maintain the expression of mesenchymal markers such as *CTNNB1* and *VIM* regardless of the cancer stage [35]. This intensified EMT program may be influenced by the changing hormonal milieu of aging, where declining androgen levels and a relative increase in estrogen could activate alternative steroid-driven programs mediated by estrogen receptors (ERs), particularly ERα, which is expressed in the prostate and has been implicated in promoting cellular proliferation [36].

### 4.2. The Tumor Microenvironment (TME)

The tumor microenvironment (TME) is a complex ecosystem of non-cancerous cells that dynamically influences tumor behavior and therapeutic response [37]. With advancing age, the systemic immune system undergoes a natural decline known as immunosenescence, resulting in immune dysfunction and an increased risk of malignancy [38]. This process directly impacts the TME and provides a compelling explanation for age-related disparities in therapeutic outcomes. In older patients, the TME is characterized by a shift toward an immunosuppressive phenotype [37]. PC is often considered an immunologically “cold” tumor, and this includes a metabolic adaptation of immune cells, with a shift from pro-inflammatory M1-like macrophages to tumor-promoting M2-like macrophages and an increased presence of myeloid-derived suppressor cells (MDSCs) [37]. The interaction between tumor cells and stromal components, particularly cancer-associated fibroblasts (CAFs) and polarized M2 macrophages, is known to foster this immunosuppressive TME, promoting motility and metastatic spread [39]. This immunosuppressive milieu creates a barrier to effective anti-tumor immune responses, providing a crucial mechanistic explanation for why immunotherapies often have limited efficacy in elderly patients and can be associated with higher rates of immune-related adverse events [38,40].

In contrast, the TME of EO-PC is characterized by a higher infiltration of anti-tumor immune cells like NK cells and dendritic cells [15]. However, this is paradoxically accompanied by the high expression of immunomodulatory genes like *CTLA4* and *IDO1*, which are associated with biochemical recurrence (BCR) [41]. This presents a state of “frustrated immunity” in younger patients, where the immune system attempts to mount a response but is actively suppressed by the tumor, leading to a more aggressive course of the disease [41].

### 4.3. The Notch Axis: A Central Integrator of Age-Dependent Phenotypes

The Notch signaling pathway is a pleiotropic cell–cell communication mechanism that presents a paradox in PC research, with conflicting reports of its function as either an oncogene or a tumor suppressor. Recent work provides a crucial resolution to this controversy by stratifying the pathway’s activity according to patient age and disease stage [35]. In younger males (under 55), Notch signaling is intensely activated, particularly at the late stage of the disease [35,42,43]. This intense activation is associated with the acquisition of a unique hybrid Sender/Receiver (S/R) phenotype by the cancer cells, characterized by the co-expression of both Notch ligands (*DLL1*, *DLL3*, *JAG1*) and receptors simultaneously. This is a departure from the traditional understanding of Notch signaling, which suggests a simple toggle switch where one cell becomes a “Sender” (high ligand, low receptor) and the other a “Receiver” (low ligand, high receptor) [35,44]. This hybrid phenotype is directly coupled with the partial/hybrid EMT program observed in younger tumors. It is associated with stem-like properties, endowing cancer cells with both epithelial (*CDH1*) and mesenchymal (*CTNNB1*, *FN1*, *VIM*) characteristics [35].

In a stark contrast, Notch signaling is found to be diminished at the early stage and inactivated at the late stage of disease in older men (over 70). The study concluded that diminished Notch signaling in elderly men worsens the prognosis. In contrast, its intense activation in younger men pre-determines a more aggressive form of PC, thus resolving the long-standing Notch paradox [35].

## 5. Clinical Implications and Therapeutic Strategies

### 5.1. Risk Stratification and Clinical Management

The unique clinical and molecular profile of EO-PC necessitates a re-evaluation of current clinical guidelines. The standard management for many low-risk PC cases is active surveillance [12]. However, this strategy may not be appropriate for young men. The evidence reveals that for carriers of *BRCA1/2* variants, radical treatment is the preferred option over active surveillance due to a higher chance of the tumor being upgraded on re-biopsy and a significantly greater risk of metastatic recurrence within 10 years [9].

This finding suggests that the traditional reliance on the initial Gleason score alone for risk stratification is insufficient for young men [45]. Given the high likelihood of a strong genetic component in this age group, a patient’s genetic profile is a more reliable indicator of disease potential than the initial clinical markers alone [9]. This underscores the critical need for a new set of clinical guidelines tailored to the genetic underpinnings of the disease in young men, where genetic testing, rather than just clinical markers, guides the decision between active surveillance and immediate radical therapy. The advent of molecular classifiers like the Decipher genomic classifier provides a crucial, complementary layer of information, refining risk stratification by assessing the risk of metastasis, BCR, and PC-specific mortality. This test helps to distinguish truly indolent tumors from those with a molecular signature of aggression despite a low Gleason score [45].

### 5.2. Personalized Medicine and Targeted Therapy

The molecular findings in EO-PC provide key therapeutic targets for a personalized medicine approach. The most significant therapeutic implication is the use of poly (ADP-ribose) polymerase inhibitors (PARPis) for patients with DNA repair pathway mutations. Studies such as the PROFOUND and TRITON-2 trials have demonstrated that PARPi, including olaparib and rucaparib, significantly improve survival and progression-free survival in patients with metastatic castration-resistant prostate cancer (mCRPC) who harbor pathogenic variants in homologous recombination repair (HR) genes, such as *BRCA1/2* [9].

Furthermore, the PROREPAIR-B study suggested that mCRPC patients with *BRCA2* pathogenic variants had a more prolonged progression-free survival (PFS) when treated with first-line abiraterone/enzalutamide (18.9 months) compared to taxanes (8.6 months), indicating a differential response to standard therapies [9]. These findings highlight how a patient’s genetic makeup can guide the selection of not only targeted therapies but also conventional treatments. The identification of distinct mutations in genes like *CDK12* and *ERF* in EO-PC further suggests that therapeutic targeting of these specific genetic drivers could be a helpful strategy in future clinical studies, opening new avenues for treatment development [26].

The profound molecular differences also necessitate a fundamental shift from a standard approach to an age-aware precision medicine framework. In older patients, the TME is predominantly immunosuppressive due to systemic immunosenescence, with an abundance of tumor-promoting macrophages and MDSCs that create a barrier to effective anti-tumor immune responses [37]. This provides a mechanistic explanation for the limited efficacy of immunotherapies in the elderly; the restricted efficacy of immune checkpoint inhibitors (ICIs) in PC is largely attributed to low tumor immunogenicity, sparse tumor-infiltrating lymphocytes, and the suppressive microenvironment characteristic of this immunologically “cold” tumor [46]. Conversely, the intense activation of Notch in younger patients, along with its association with a highly plastic and aggressive phenotype, suggests that targeting this pathway with gamma-secretase inhibitors (GSIs) may be particularly effective in this cohort [35,43]. Preclinical studies have demonstrated that therapeutic inhibition of Notch with GSIs can decrease proliferation, invasion, and tumorsphere formation, and that this effect can be synergistically enhanced with anti-androgen therapies or chemotherapy like docetaxel [47]. The fact that Notch is inactivated in older patients suggests that a different therapeutic strategy is needed for this group [35]. This provides a potential explanation for the disappointing clinical performance of GSIs in many solid tumors; the trials may have failed because they did not properly stratify patients based on their Notch activation status, a crucial distinction now provided by this age-delineated framework [47].

## 6. Future Directions and Knowledge Gaps

Existing reports provide conflicting views regarding the clinical characteristics and prognosis of PC in young men, highlighting a significant knowledge gap in this area [12]. There is a critical need to establish specific, evidence-based clinical guidelines tailored to this patient population. These guidelines must integrate genetic risk factors and account for the unique paradox of the disease’s presentation, moving beyond the standard approach that currently dominates clinical practice.

Further research is required to fully characterize the molecular heterogeneity of EO-PC. While some key mutations have been identified, a more complete understanding of the functional roles of newly identified mutations like *CDK12* and *ERF* is needed. This includes not only their mechanistic role in oncogenesis but also the development of specific, effective therapies to target them [26]. As recapped in Table 2, the distinct transcriptional and epigenetic profiles of early-onset tumors, characterized by a unique inflammatory and immune signature, also warrant further investigation. A deeper understanding of these pathways could lead to the development of novel biomarkers for early detection and targeted immunotherapies [41].

## 7. Conclusions

The evidence overwhelmingly supports the conclusion that PC is not a single disease. Patient age serves as a fundamental stratifying variable, delineating two distinct molecular and clinical entities, i.e., EO-PC and LO-PC. EO-PC is characterized by unique fusion events [18], a highly plastic partial EMT driven by intense Notch activation [35], and an inflammatory yet immunosuppressed TME [41]. In contrast, LO-PC is defined by a higher mutational burden, an epigenetic “field defect” of aging [24], a different, multi-factor-driven EMT program [34,35], and a predominantly immunosuppressive TME with inactivated Notch signaling (Table 2) [37].

These biological differences have profound clinical implications. Traditional prognostic tools, such as the Gleason score, are insufficient to capture the full spectrum of risk, as demonstrated by the surprisingly high rate of BCR in younger patients [41,51]. The effectiveness of specific therapies, particularly immunotherapies, may be reduced in older patients due to systemic immunosenescence [38]. The future of PC management lies in the development of an age-aware precision medicine framework that integrates multi-omic data, functional platforms, and computational models to predict disease progression and optimize treatment. This new paradigm promises to improve prognostic stratification, guide personalized therapeutic decisions, and ultimately lead to better clinical outcomes for all men affected by this complex disease.

## Figures and Tables

**Table 1 ijms-26-09777-t001:** Comparative clinical and pathological features of EO-PC vs. LO-PC.

Characteristic	EO-PC ≤ 55 Years	LO-PC ≥ 65 Years
Incidence	Rising, over 10% of new US diagnoses [11].	In the majority of cases, two-thirds are diagnosed in people over age 70 [9].
5-year relative survival	98% (for ages 20–54) [11].	100% (for ages 55–79) [11].
Disease characteristics	The majority are low-risk at diagnosis; however, a subset with high-grade/stage has a poor prognosis [11].	The majority are indolent, but age is a significant risk factor for all grades and stages [11].
Cause-specific mortality	Among high-grade/stage cases, higher mortality than in all other groups except those over 80 [11].	Lower mortality than young men in similar high-grade/stage cases [11].
Genetic component	More significant genetic component, linked to constitutional variants [9].	A less significant heritable component, more driven by age and environmental factors [9].
Therapeutic burden	Prolonged impact from treatment-related morbidities due to extended life expectancy [11].	Shorter duration of long-term morbidities [11].
Paternal disease characteristics	A father diagnosed at <65 years is associated with a >4-fold increase in risk; a father with a Gleason score ≥8 is associated with a >2.5-fold increase in the son’s risk for a similar high-grade cancer [10].	Risk is significantly lower with a father diagnosed at ≥70 years old (SIR, 2.3 vs. 4.3 for <65 years) [10].

**Table 2 ijms-26-09777-t002:** Key molecular differences between EO-PC and LO-PC.

Characteristic	EO-PC ≤ 55 Years	LO-PC ≥ 65 Years
Genomics	Enriched for fusion events (e.g., *TMPRSS2-ERG*) driven by chromoplexy; lower mutational burden [18,19].	Higher overall mutational burden, with a “clock-like” C>T signature; enriched for mutations in tumor suppressors (*RB1*, *APC*, *AR*) [24,25].
Epigenomics	Less accumulation of AREAS [29].	Widespread epigenetic drift; accumulation of AREAS with hypermethylation of genes like *GSTP1* and *RASSF1A* [23,24].
Transcriptomics	Higher neuroendocrine differentiation and MAPK pathway activity scores [17].	General AR- and ERα-linked programs [48].
EMT profile	Partial/Hybrid EMT; co-expression of epithelial (*CDH1*) and mesenchymal markers (*CTNNB1*) [34,35].	Multifactor-driven EMT [34,49,50].
TME	“Frustrated immunity” with high immune cell infiltration (NK, DC) but also high expression of immunosuppressive genes (*CTLA4*, *IDO1*) [41].	Immunosuppressive TME with a shift to M2-like TAMs and increased MDSCs [37].
Notch signaling	Intense activation at the late stage, leading to a hybrid Sender/Receiver phenotype [35].	Diminished at the early stage, inactivated at the late stage [35].
Prognosis	High rate of BCR, despite low Gleason score [41,51].	Lower recurrence for the same clinical stage; prognosis often tied to comorbidities [1].

## Data Availability

Not applicable.

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
