# Peer review of "Molecular Landscape of Prostate Cancer Across Age Groups: Impact on Prognosis and Treatment Outcomes"

_ijms, 2025, doi:10.3390/ijms26199777_

Round 1
Reviewer 1 Report
Comments and Suggestions for Authors
In the review article titled “Molecular Landscape of Prostate Cancer Across Age Groups: Impact on Prognosis and Treatment Outcomes ” authors present a comprehensive analysis of the clinical, pathological, and molecular characteristics of PC. Even though the authors have thoroughly reviewed the literature around Early Onset PC there is limited text on the Late Onset PC in the main text.
1. Line 32: Authors should refer to Cancer Statistics, Siegel et.al . (2025) which is a comprehensive review
2. Line 110: Authors discuss the effect of PC diagnosis of father and son, which establishes the ground for inheritance. Authors should conclude the chapter by discussing their point of view on diagnosis or the fact that PC is genetic.
3. Section 4.2: Here the authors discuss the tumor micro environment, but as PC is know to be immunologically cold, it is important to refer to such studies and discuss the impact of immune therapy.
4. Even though the authors state that “This review provides a comprehensive analysis of the clinical, pathological, and 69 molecular differences that distinguish early-onset (EO-PC) PC from its late-onset (LO-PC) 70 counterpart” there is no clear comparison between the types of PC in the main text. Authors should consider rephrasing the statement.
Author Response
Comments 1: Line 32: Authors should refer to Cancer Statistics, Siegel et.al. (2025) which is a comprehensive review
Response 1: Thank you for pointing this out. We agree with this comment, as citing the most current epidemiological data is crucial. Therefore, we have added a specific reference to Siegel et al. (2025) and incorporated a sentence highlighting the significant racial disparities in PC mortality, as detailed in that comprehensive review. This change is located in Introduction.
"As highlighted in the comprehensive Cancer Statistics review by Siegel et al. (2025), significant racial disparities in PC mortality persist; for instance, the mortality rate among Black men (36.9 deaths per 100,000) is more than double that of White men (18.4 deaths per 100,000)."
Comments 2: Line 110: Authors discuss the effect of PC diagnosis of father and son, which establishes the ground for inheritance. Authors should conclude the chapter by discussing their point of view on diagnosis or the fact that PC is genetic.
Response 2: We agree that the conclusion of this section required a stronger statement regarding the clinical implication of the genetic findings. We have added a concluding sentence to Section 2.2 that emphasizes the pivotal role of genetic testing in risk stratification for EO-PC, which is clearly rooted in inherited factors.
"Given that EO-PC is deeply rooted in genetic factors, genetic testing becomes a pivotal diagnostic component, enabling the assessment of the true risk of disease progression and guiding more personalized therapeutic decisions, even in cases with a seemingly benign initial clinical picture."
Comments 3: Section 4.2: Here the authors discuss the tumor micro environment, but as PC is know to be immunologically cold, it is important to refer to such studies and discuss the impact of immune therapy.
Response 3: Thank you for this critical suggestion. We have revised both the descriptive section on TME (Section 4.2) and the therapeutic implications section (Section 5.2) to clearly establish the immunologically "cold" nature of PC and its impact on immunotherapy efficacy in the LO-PC cohort.
Change 1 (Section 4.2): The TME description is enhanced with the context of PC being a cold tumor and the role of stromal interactions.
"The TME in LO-PC is characterized by a shift toward an immunosuppressive phenotype. This includes a metabolic adaptation of immune cells, with a shift from pro-inflammatory M1-like macrophages to tumor-promoting M2-like macrophages and an increased presence of myeloid-derived suppressor cells (MDSCs). Prostate cancer is often considered an immunologically "cold" tumor, and this immunosuppressive milieu creates a critical barrier to effective anti-tumor immune responses, explaining why immunotherapies often have limited efficacy in elderly patients. The interaction between tumor cells and stromal components, particularly cancer-associated fibroblasts (CAFs) and polarized M2 macrophages, is known to foster this immunosuppressive TME, promoting motility and metastatic spread."
Change 2 (Section 5.2): The discussion on therapeutic limitations is strengthened by referencing low immunogenicity.
"In older patients, the TME is predominantly immunosuppressive due to systemic immunosenescence, with an abundance of tumor-promoting macrophages and MDSCs that create a barrier to effective anti-tumor immune responses. This provides a mechanistic explanation for the limited efficacy of immunotherapies in the elderly; the restricted efficacy of immune checkpoint inhibitors (ICIs) in PC is largely attributed to low tumor immunogenicity, sparse tumor-infiltrating lymphocytes, and the suppressive microenvironment characteristic of this immunologically "cold" tumor."
Comments 4: Even though the authors state that “This review provides a comprehensive analysis of the clinical, pathological, and 69 molecular differences that distinguish early-onset (EO-PC) PC from its late-onset (LO-PC) 70 counterpart” there is no clear comparison between the types of PC in the main text. Authors should consider rephrasing the statement.
Response 4: We agree that the initial statement was too broad given the detailed comparative nature of our analysis (especially the tables and the sectional breakdown). We have rephrased the sentence in Introduction, to reflect that the review provides a direct comparison of the divergent trajectories, addressing the reviewer's concern about the lack of "clear comparison."
"This review details the divergent molecular and clinical trajectories that distinguish EO-PC from its late-onset counterpart, providing a direct comparison necessary for developing age-aware precision medicine strategies."
Reviewer 2 Report
Comments and Suggestions for Authors
The authors focused on early-onset prostate cancer (EO-PC), which is typically defined as a diagnosis before the age of 55, and compared it to late-onset (LO-PC), which is typically defined as a diagnosis over the age of 65. Interestingly, a comparison of EO-PC and LO-PC reveals distinguishing characteristics, with EO-PC exhibiting a pronounced familial genetic component, distinctive fusion events (e.g., TMPRSS2-ERG), a high frequency of CDK12 and ERF mutations, and activation of the Notch signaling pathway. The findings of this study indicate the necessity for a distinct therapeutic approach for EO-PC, one that differs from the standard treatment methods employed for LO-PC. This review is well-written and highly informative. There were some minor questions ad suggestions as described below.
Major comment
- The definitions and differences between EO-PC and LO-PC are important in this review, but the authors provide their definitions on page 5, lines 184 and 204, respectively. They should have included these definitions earlier to make them easier to understand.
Minor comments
- What means the words “age 6.5” on page 3, line 93? Is it a mistake for "age 65"?
- The authors should remove “1” from “3.8-5.0).1” on page 3, line 108.
- “Early-onset PC” should changed to “EO-PC” on page 5, line 184.
- The delineation between the various components of the Table is not readily apparent and is a source of confusion. The authors should correct it.
Author Response
Comments 1: The definitions and differences between EO-PC and LO-PC are important in this review, but the authors provide their definitions on page 5, lines 184 and 204, respectively. They should have included these definitions earlier to make them easier to understand.
Response 1: Thank you for this major comment. We completely agree that the definitions of EO-PC and LO-PC must be presented immediately to establish the framework of the review. We have implemented this major structural change by moving the definitions from Section 2 (original location) to Introduction, second paragraph, ensuring they are established early in the manuscript.
The revised text now reads:
"In this report, Early-Onset Prostate Cancer (EO-PC) is defined as a diagnosis established at age ≤55 years, while Late-Onset Prostate Cancer (LO-PC) includes men diagnosed at age ≥65 years. The above age distinction is not arbitrary but reflects fundamentally different evolutionary pathways of the tumor."
Comments 2: The delineation between the various components of the Table is not readily apparent and is a source of confusion. The authors should correct it.
Response 2: Thank you for this important note regarding clarity. We agree that simplifying the column headers enhances readability, especially given the density of comparative data. We have corrected this issue in Table 1 and Table 2 by simplifying the column headers.
Table 1: Comparative clinical and pathological features of EO-PC vs. LO-PC.
The headers for the second and third columns were changed from EO-PC1 and LO-PC2 to EO-PC <55 years and LO-PC >65 years, respectively.
Table 2: Key molecular differences between EO-PC and LO-PC.
The headers for the second and third columns were changed from EO-PC1 and LO-PC2 to EO-PC <55 years and LO-PC >65 years, respectively.
The remaining minor comments have all been included, and the manuscript has been modified accordingly.